# SARS-CoV-2 Genetic Variants Identified in Selected Regions of Ethiopia Through Whole Genome Sequencing: Insights from the Fifth Wave of COVID-19

**DOI:** 10.3390/genes16030351

**Published:** 2025-03-18

**Authors:** Getnet Hailu, Mengistu Legesse, Andargachew Mulu, Girmay Medhin, Mesfin Mengesha Tsegaye, Dawit Hailu Alemayehu, Abaysew Ayele, Atsbeha Gebreegziabxier, Adamu Tayachew, Adimkewu Aguine, Haileyesus Dejene, Sofonias K. Tessema, Harris Onywera, Assohoun Egomli Stanislas, Ebba Abate, Alessandro Marcello, Molalegne Bitew

**Affiliations:** 1Aklilu Lemma Institute of Pathobiology, Addis Ababa University, Addis Ababa P.O. Box 1176, Ethiopia; getnethailu21@gmail.com (G.H.); mengistu.legessed@aau.edu.et (M.L.); gtmedhin@yahoo.com (G.M.); adamutayachew@gmail.com (A.T.); ebbaabate@yahoo.com (E.A.); 2Ethiopian Public Health Institute, Addis Ababa P.O. Box 1242, Ethiopia; atsbehag@gmail.com (A.G.); adime12ag21@gmail.com (A.A.); 3Armaur Hansson Research Institute, Addis Ababa P.O. Box 1005, Ethiopia; andargachew.mulu@ahri.gov.et (A.M.); mesfintiru@gmail.com (M.M.T.); dawit.hailu@ahri.gov.et (D.H.A.); abaysew.ayele@ahri.gov.et (A.A.); 4College of Veterinary and Animal Science, University of Gondar, Gondar P.O. Box 196, Ethiopia; haileyesushdejene@uog.gov.et; 5Africa Centres for Disease Control and Prevention (Africa CDC), Addis Ababa P.O. Box 3243, Ethiopia; sofoniast@africa-union.org (S.K.T.); do79harris@gmail.com (H.O.); 6Genomics and Metagenomics Platform, Pasteur Institute, Abidjan P.O. Box 75015, Côte d’Ivoire; stanlasso@gmail.com; 7Laboratory of Molecular Virology, International Centre for Genetic Engineering and Biotechnology (ICGEB), Padriciano 99, 34149 Trieste, Italy; marcello@icgeb.org; 8Bio and Emerging Technology Institute, Addis Ababa P.O. Box 5954, Ethiopia

**Keywords:** SARS-CoV-2, genetic divergence, whole genome sequencing, fifth wave, Ethiopia

## Abstract

Background: The COVID-19 pandemic highlighted SARS-CoV-2 variants with increased transmissibility and immune evasion. In Ethiopia, where cases surged, the understanding of the virus’s dynamics was limited. This study analyzed SARS-CoV-2 variants during the fifth wave, crucial for guiding vaccines, therapeutics, diagnostics, and understanding disease severity. Method: From June to August 2022, 150 SARS-CoV-2-positive samples were randomly selected from the Ethiopian Public Health Institute repository. Sixty-three high-quality genome sequences were analyzed. Results: Of the 63 sequences, 70% were from males and 30% from females, with a median age of 34. Omicron dominated (97%, 61/63), primarily clade 22A (64%, 40/63), followed by 22B (18%, 11/63) and 21K (14%, 9/63). Delta accounted for 3.2% (2/63). Omicron was identified in all (25) vaccinated study participants. Ethiopian sequences showed limited evolutionary divergence and lower genetic diversity compared to global sequences. Conclusion: Omicron was the predominant variant during Ethiopia’s fifth wave, indicating recent community transmission. Despite minor genetic diversity differences, ongoing surveillance remains critical for tracking variants and informing public health interventions.

## 1. Introduction

Coronavirus disease (COVID-19) is an infectious disease caused by the severe acute respiratory syndrome coronavirus 2 (SARS-CoV-2) which originated in Wuhan City, Hubei Province, in China [1]. SARS-CoV-2 is transmitted from person to person through the inhalation of respiratory droplets from an infected individual or through direct contact with infected droplets through the eyes, mouth, or nose. However, transmissibility, incubation period, and infectivity vary depending on the variants and individual immunity [2].

Coronaviruses are enveloped RNA viruses belonging to the *Coronaviridae* family, characterized by a positive single-stranded genome exceeding 29,891 nucleotides, encoding for 9890 amino acids [3]. Their genome includes numerous open reading frames (ORFs) that produce structural (SPs) and non-structural proteins (NSPs) essential for the virus’s life cycle and pathogenesis [4]. Structural proteins include the spike (S), envelope (E), membrane (M), and nucleocapsid (N) proteins, while non-structural proteins (nsp1–16) facilitate viral metabolism and interaction with the host immune system [5]. The spike glycoprotein of SARS-CoV-2 is cleaved into two subunits, S1 and S2, facilitating virus–cell membrane fusion [6]. The S1 subunit contains receptor-binding and N-terminal domains, crucial for viral entry and potential targets for neutralization. Host protease primes the S2 subunit [6]. Viral mutations can lead to new pathogenic variants affecting transmission, disease severity, and vaccine efficacy [7].

The COVID-19 pandemic was first reported in Africa on 14 February 2020, with the initial confirmed case detected in Egypt [8]. Ethiopia confirmed its first COVID-19 case in March 2020. The wide spread of COVID-19 led the World Health Organization (WHO) to declare the disease as a pandemic [9]. On 8 April 2020, the national government of Ethiopia declared a five-month state of emergency but allowed economic activities to continue during the public health crisis [10]. Ethiopia has experienced multiple pandemic waves, similar to the global trend. The first wave occurred from July to December 2020, followed by the second wave from January to July 2021. The third wave was recorded between July and December 2021, while the fourth wave spanned from December 2021 to April 2022. The fifth wave then emerged from June to August 2022. As of April 2024, from the total of more than 4 million tested, 501,157 positive cases were reported. From these, 488,171 and 7574 (%) individuals recovered and died, respectively. And only 38% of the population received at least one dose of a COVID-19 vaccine [11].

A sero-epidemiological survey conducted in Addis Ababa and Jimma between August 2020 and April 2022 revealed that over 96% of the study group had been exposed to SARS-CoV-2 at least once [12]. This figure indicates a high level of exposure across the population. Similarly, most Ethiopians have had multiple exposures to SARS-CoV-2, leading to high antibody titers with slow decay characteristics. This suggests the development of hybrid immunity due to recurrent infections with different variants and vaccination among many individuals [13].

At the beginning of the pandemic, Ethiopia, like many countries, primarily dealt with the initial SARS-CoV-2 strains. The virus’s rapid spread and the emergence of variants were closely monitored as part of global efforts to understand the virus’s behavior and adapt public health measures. As the virus continued to circulate, variants began to emerge. Variants of concern (VOCs) like Alpha (B.1.1.7), Beta (B.1.351), Gamma (P.1), Delta (B.1.617.2), and Omicron (B.1.1.529) have been identified in Africa [14,15]. In Ethiopia, similar variants have likely been detected [16]. Unfortunately, SARS-CoV-2 in the country was not sequenced during the actual pandemic period. This was due to a lack of laboratory infrastructure, skilled human resources, and reagents and kits. However, due to the relative improvement of sequencing infrastructure in the country, this project sequenced 63 retrospective samples from the national repository of the Ethiopian Public Health Institute collected during the fifth pandemic wave.

The dynamics of SARS-CoV-2 variants were not consistently monitored as part of the national surveillance systems in Ethiopia. This information is crucial for informing public health interventions, monitoring the genomic epidemiology of the virus, identifying specific regional challenges, and supporting research and collaboration [13]. Only 824 SARS-CoV-2 samples were sequenced and submitted to GISAID from Ethiopia to date. This is relatively very small compared to other African countries like South Africa (56,886), Kenya (13,770), Egypt (5146), and Tunisia (2915). Out of 824 SARS-CoV-2 samples sequenced and submitted, 63 were for this project. This shows how SARS-CoV-2 sequencing activities were not conducted during the actual outbreak period. So, any information generated in this manuscript is very novel for the Ethiopian condition. The global science community can get information about what was the situation of SARS-CoV-2 variants in Ethiopia during the 5th wave of the pandemic from this manuscript as well as from the GISAID/NCBI submissions we made. This was the main rationale for initiating this study project. Therefore, this study was conducted to investigate the SARS-CoV-2 variants that circulated during the fifth wave of the pandemic in selected regions of Ethiopia.

## 2. Materials and Methods

### 2.1. Study Design, Period, and Sample Collection

A retrospective experimental study design was used. The study design was a retrospective experimental design. Sample collection was undertaken during the fifth wave of the pandemic (June to August, 2022) in Ethiopia. A total of 150 SARS-CoV-2-positive samples were collected retrospectively from the national virology reference laboratory repository at the Ethiopian Public Health Institute (EPHI). These samples were nasopharyngeal (NP) swab samples collected from patients residing in Addis Ababa, Oromia, Amhara, and Southern Nation Nationalities and Peoples (SNNP) regions (Figure 1). EPHI is a national reference laboratory for SARS-CoV-2 in Ethiopia, and samples were brought from those geographic areas, as well as from diplomatic communities and international travelers, for viral testing. Samples were stored at −80 °C for long-term storage. Samples for sequencing were purposively selected based on their identifiers (IDs) from the electronic national database. Only the IDs that tested positive and had a cycle threshold (Ct) value less than or equal to 30 were selected. As a result, 150 SARS-CoV-2-positive samples were included in this study due to our limited sequencing resources. The patients’ metadata, including demographic information, clinical data, and RT-qPCR results, were anonymously extracted from electronic databases. The study participants were then grouped based on factors such as age, sex, signs and symptoms, pre-existing chronic diseases, travel history, and vaccination status.

### 2.2. Quality Assurance

To increase the success rate of whole genome sequencing, an RT-qPCR test was performed again after the samples were thawed at room temperature [17]. The RNA from these samples was re-extracted using the BGI extraction kit, and detection was performed using a BIO-RAD CFX96 Deep Well™ Real-Time PCR Detection system (Bio Rad Laboratories, Inc., Hercules, CA, USA) following the standard protocol. The clinical samples were handled and processed in a Biosafety Laboratory-2 (BSL-2) at the national virology reference laboratory. Out of the 150 re-analyzed samples, 70 had a Ct value less than or equal to 30 and were selected for whole genome sequencing, and others were discarded due to their high Ct values (i.e., >30). The RNA extracts from these 70 samples were sequenced using the Illumina NextSeq 550 platform, San Diego, CA, USA, at the virology laboratory of EPHI during March 2023. Out of 70 SARS-CoV-2 sequence data, 63 sequences passed the bioinformatics quality controls requirements (i.e., with genome coverage greater than 80%, no clustered mutations, and no misplaced stop codons) and were selected for further sequence analysis.

### 2.3. RNA Extraction and qRT-PCR

Total RNA was extracted from 300 μL nasopharyngeal samples using MegaBio plus Virus Purification Kit II (BSC87S1E), (Hangzhou Bioer Technology Co., Ltd., Hangzhou, China), using the Bioner Gene Pure Pro fully automatic Nucleic Acid Purification System (NPA-32P) per the manufacturer’s instructions [18]. Real-time qRT-PCR was performed to confirm the presence of the virus in the samples using primers and probes corresponding to SARS-CoV-2 with the QIAamp Viral RNA Mini Kit (Qiagen, Shanghai, China) according to the manufacturer’s protocol [19]. qRT-PCR was performed using the BIO-RAD CFX96 Deep Well™ Real-Time PCR Detection system (Bio-Rad Laboratories, Inc., USA). All samples were tested for the human RNase P gene as an internal control.

### 2.4. Complementary DNA (cDNA) Synthesis, DNA Library Preparation, and Whole Genome Sequencing

Seventy RNA extracts, which exhibited low Ct values (Ct < 30) in quantitative real-time PCR assays, were chosen for library preparation. The cDNA synthesis, SARS-CoV-2 sequence enrichment, library amplification, and indexing were performed using the Illumina COVIDSeq test (RUO) (Illumina, USA) following the manufacturer’s instructions. The resulting libraries were pooled, normalized, and quantified using a Qubit^®^ 4 fluorometer (Thermo Fisher Scientific, Waltham, MA, USA) [20] before paired-end sequencing was performed on a NextSeq system (Illumina, USA) with a NextiSeq 500/550 cycle V3 kit (Illumina, USA).

### 2.5. Sequence and Metadata Analysis

The metadata were exported from the national database and converted into Excel format. The data were encoded into STATA v17, and descriptive analyses were performed. Raw sequencing data were obtained in FASTQ format and processed for variant calling and consensus sequence generation. First, the FASTQ sequences’ quality was evaluated by the FastQC (v0.11.9) program, and then adapter sequences were trimmed from reads using Cutadapt software v.4.6 [21] to remove adaptor sequences and poor-quality reads. An average base quality of Q30 was used for trimming. The trimmed reads were then mapped against the SARS-CoV-2 WuhanHu-1 reference genome sequence (GenBank accession No. NC_045512.2) using the Burrows–Wheeler Alignment BWA mem v0.7.12 [22]. The consensus sequence was then generated using Samtools version 1.12. Moreover, SARS-CoV-2 variants were called with locally installed Pangolin (daily updated version [23], and mutations were identified with online Nextclade v3.8.2 Clade assignment, mutation calling, and sequence quality checks [24]. All viral genome sequences obtained from this study were submitted to the Global Initiative on Sharing All Influenza Data (GISAID) with accession IDs from EPI_ISL_19,147,618 to EPI_ISL_19302657 [25] and National Center for Biotechnology Information (NCBI) repositories with accession numbers from PQ140559 to PP069552 [26].

### 2.6. Computation of SNP Distances in Samples to the SARS-CoV-2 Reference Genome

SNP-dist v0.70 [27] was used to determine the number of SNPs in the 63 sequences compared to the SARS-CoV-2 reference genome. The hCoV-19/Wuhan/WIV04/2019 sequence from Wuhan, China, which consists of 29,891 base pairs, was used as a reference genome obtained from the GISAID. The snp-dist tool calculates the SNP distance matrix from a dataset of multiple sequence alignments (MSAs), where the sequences are of the same length. These distance matrices are commonly used in studies of infectious disease outbreaks. For this analysis, we only considered informative bases (A, T, C, and G) and not Ns. MSA was conducted using MAFFT v7.526 [28], employing the multiple alignment technique with the Fast Fourier Transform. The following parameters were applied: a gap opening penalty of 1.53 for group-to-group alignment and a gap extension penalty of 0.00 for group-to-group alignment. The block substitution matrix (BLOSUM) was selected, with a coefficient of 62, and the L-INS-I alignment method of MAFFT was chosen, known for its accuracy in alignments of ≤200 sequences.

### 2.7. Evolutionary Divergence Between Sequences and Overall Sequence Pairs

To estimate the evolutionary divergence between sequences as well as the average evolutionary divergence over all sequence pairs, MSA of the 63 sequences was performed using MAFFT v7.526 [28]. The output of the alignment was used to construct the phylogenetic tree via IQ-tree version v2.3.5-3 [29]. All ambiguous positions were removed for each sequence pair (pairwise deletion option). There was a total of 29,921 positions in the final dataset. The pairwise and overall mean (average) distances were used as proxies for evolutionary divergence.

For comparison purposes, the overall evolutionary divergence among all sequence pairs obtained from the GISAID uploaded during and before the study period was calculated. This analysis included the WIV04 genome and a total of 118 SARS-CoV-2 genome sequences from 62 countries. The sequences encompassed various variants, including former VOCs present in neighboring countries and different regions worldwide (Table 1).

### 2.8. Phylogenetic Analysis of SARS-CoV-2 Genome Sequences

The IQ-TREE v2.3.5.3 was used to compute the phylogenetic inference of the 63 SARS-CoV-2 genome sequences [29]. In addition to these 63 sequences, after the consensus sequences obtained in this research were aligned, the top 118 SARS-CoV-2 genomes were selected and downloaded from GISAID for phylogenetic analysis. First, an alignment analysis using MAFFT v7.526 was performed [28]. The MSA which had 30,058 nucleotide sites was then imported to IQ-TREE. The phylogeny involved determining the best-fit DNA substitution model using Model Finder [6], which tested 88 DNA models. The initial parsimony tree was created by the phylogenetic likelihood library (PLL). The model chosen according to the Bayesian Information Criterion (BIC) score (115,726.5435) was the TIM + F + I + G4. The model of rate heterogeneity was as follows: proportion of invariable sites: 0.8736 and Gamma shape alpha: 0.9901 with 4 categories. The rate parameter (R) was as follows: A-C: 1.0000, A-G: 1.6270, A-T: 0.5506, C-G: 0.5506, C-T: 3.9021, and G-T: 1.0000. A bootstrap analysis using the UFBoot2 method with 1000 replicates and ML heuristic method was performed. Additionally, branch lengths were optimized through ML on the original alignment. The initial trees for the heuristic search were generated using BioNJ algorithms. From these analyses, we selected the phylogenetic tree with the highest log-likelihood value (−55,910.033). The resulting phylogenetic trees were saved in Newick format and visualized using the Interactive Tree of Life interface [30]. Statistical analysis of the demographic and clinical information was performed using STATA v17. Frequency distribution and tabulation were performed to show the relationship between outcome and independent variables.

## 3. Results

### 3.1. Demographics and Clinical Characteristics of the Study Subjects

Out of 150 samples collected, 70 samples had a Ct value of less than 30 and were sequenced. Out of 70 samples sequenced, 63 samples passed the bioinformatics quality control. Of the 63 genome sequences obtained, the samples were collected from patients in the age range of 14 to 75 years, with a median age of 34 years. Of the study participants, 32% (20/63) belong in the age group 21–30 years old, followed by 21% (13/60) in 31–40 years and 16% (10/63) in 41–50 years old. Females accounted for 30% (19/63) and males for 70% (44/63). The majority of the sequenced samples were collected from Addis Ababa (70%; 44/63), followed by Oromia (24%; 15/6), Amhara (5%; 3/63), and Southern Nation Nationalities Region (SNNPR) (2%; 1/63). In terms of clinical data, 70% (44/63) of the study participants exhibited signs and symptoms of cough, fever, and anosmia, and 43% (27/63) of participants reported they had known chronic disease. In addition, 13% (8/63) had a travel history outside Ethiopia, delayed 8–11 days before testing, and approximately 40% (25/63) reported having a vaccination history (Table 2).

### 3.2. Sequencing and Sequence Analysis

The majority of the sequences (99.8%) had high quality, with gap sizes ranging from 20 to 114 nt, accounting for approximately 0.4% of the total sequence. On average, each genome had 69.21 single nucleotide variations (SNVs) in different genes of SARS-CoV-2. The highest numbers of single nucleotide mutations per clade of SARS-CoV-2 were identified in 22A and 21K (Figure 2).

Nextclade variant analysis shows that out of the 63 sequences examined, on average, 69 nucleotide mutations were identified across the samples in comparison to the Wuhan reference genome (Figure 3).

This analysis also detected an average of 198 nucleotide deletions, 68 substitutions, and 1 insertion in different genes of SARS-CoV-2 (Appendix A). These SARS-CoV-2 nucleotide mutations were observed in different positions of S, M, and N genes and genes coding for non-structural proteins (nsp1, nsp6, nsp9, nesp10, nsp13, ORF3a, and ORF7a) (Appendix A). Among them, 371 nucleotide deletions were observed in the S gene in the positions of 25–27, 69–70, 124–144, 157–158, 212, 374–376, 415–432, 446–464, and 758–767, followed by 177 deletions in N in the position of 31–33 and 168 deletions in the nsp6 gene in the position of 105–108.

This study identified several mutations in the PLpro (papain-like protease), including K38R, V1069I, A1892T, D821N, E906A, T24I, G489S, I684V, S370L, V932L, P985S, T703I, A1179V, P822L, A488S, P1228L, and P1469S. Additionally, this study explores the presence of the P132H mutation in the 3CLpro (main protease), and P323L mutation in RdRP.

In this study, the transition/transversion bias (R) was 1.693, indicating that transitions occur more frequently than transversions. The base substitution rates (r) for each nucleotide pair were as follows: A => T (0.06), A => C (0.03), A => G (0.07), T => A (0.05), T => C (0.17), T => G (0.03), C => A (0.05), C => T (0.29), C => G (0.03), G => A (0.11), G => T (0.06), and G => C (0.03). These results show that thymine (T) has the highest substitution rate with cytosine (C) (0.17) and the lowest with guanine (G) (0.03). Cytosine (C) has the highest substitution rate with thymine (T) (0.29), while its rates with adenine (A) and guanine (G) are relatively low (both 0.03). Adenine (A) and guanine (G) have generally similar substitution rates across different nucleotides, with A => G having the highest rate among adenine substitutions and G => A having the highest rate among guanine substitutions. In this study, the average Ti/Tv ratio was 1.9734, ranging from 1.5862 to 2.4737, and the most common nucleotide change was the G > A transition.

### 3.3. SARS-CoV-2 Variants Identified

This study found that omicron was the dominant variant (97%, 61/63) identified in the total sample sequenced during the fifth wave of the pandemic in selected regions of Ethiopia, followed by Delta variants (3%, 2/63). All of the identified cases of the Omicron variant belonged to four age groups (<20, 21–30, 31–40, 41–50, 51–60, and >60 years old). In the age groups of 21–30 and 41–50 years old, Delta accounted for 5% and 10% of the identified cases, respectively, and the remaining infections were caused by Omicron variants. In addition, all of the 25 individuals who were vaccinated before testing were infected with Omicron variants. Out of the 38 non-vaccinated study participants, 95% (36/38) were infected with Omicron, and 5.3% (2/38) were identified with Delta variants (Table 3).

Almost comparable numbers of vaccinated and unvaccinated participants were infected with the BA.4.1 and BA.4.1.1 sub-lineages of SARS-CoV-2 (Figure 4). According to the Nextstrain classification, the dominant clades identified during the study period were 22A (64%, 40/63) and 22B (18%, 11/63), followed by 21K (Omicron; 14%, 9/63), 21J (Delta, 3.2%, 2/63), and 21L (Omicron; 1.6%, 1/63). Additionally, sub-lineages BA.4.1 (30.2%, 19/63) and BA.4.1.1 (28.5%, 18/63) were more prevalent than BA.1.1 (9.5%, 6/63), BA.5.2 (7.9%, 5/63), BF.2 (6.4%, 4/63), and BA.4 (4.8%, 3/63). Interestingly, 90% of the sub-lineages belonged to different strains within the BA lineage, which represents a specific genetic variant or group of closely related variants of the SARS-CoV-2 virus. Additionally, 6.4% (4/63) of the samples were part of the BF lineage, while the AY and B lineages each accounted for 1% of the samples.

### 3.4. Evolutionary Divergence and Phylogenetic Analysis of SARS-CoV-2

The evolutionary analysis among sequenced data for this study showed there is evidence of a phylogenetic relationship in some of the sequences. The evolutionary divergence between sequences ranges from 0.00 to 3.45 × 10^−3^ substitution per site. Looking at the broader analysis, the average evolutionary divergence across all sequence pairs is similarly low. For sequences from Ethiopia, the overall distance is calculated to be 9.78 × 10^−4^ substitutions per site, with a standard error of 8.47 × 10^−5^ substitutions per site. In comparison, when considering selected global sequences, the overall distance slightly increases to 1.54 × 10^−3^ substitutions per site, with a standard error of 1.19 × 10^−4^ substitutions per site. In addition, the maximum likelihood phylogenetic tree from our study, comprising 63 SARS-CoV-2 genomes, demonstrated seven main dominant clades in the viral population (Figure 5).

## 4. Discussion

COVID-19, caused by the SARS-CoV-2 virus, first emerged in Wuhan, China, before spreading globally, including to Ethiopia. SARS-CoV-2 is primarily transmitted through respiratory droplets and direct contact. The spread of the virus is influenced by various factors, including sunlight, climate, temperature, humidity, and population demographics [31]. Asymptomatic and pre-symptomatic individuals play a significant role in the transmission of the virus [32]. It possesses a large RNA genome that encodes various proteins essential for its life cycle and pathogenesis, with the spike protein being particularly important for viral entry and a key target for neutralization. Mutations in the virus can lead to the emergence of new variants that may impact transmission rates, disease severity, and vaccine efficacy [33].

In Ethiopia, the first COVID-19 case was reported in March 2020, and by April 2024, the country had recorded over 500,000 cases, with approximately 38% of the population vaccinated [34,35]. A sero-epidemiological survey indicated high exposure rates among the population [12]. While Ethiopia initially contended with early virus strains, variants of concern emerged globally, although local sequencing efforts were hampered by infrastructure challenges. This study aims to investigate the SARS-CoV-2 variants circulating during the fifth pandemic wave in selected regions of Ethiopia, addressing a critical gap in the country’s genomic surveillance capabilities. The findings from this study provided significant insights into the demographics, clinical characteristics, and SARS-CoV-2 variants circulating in selected regions of Ethiopia during the fifth wave of the pandemic.

In this study, the age distribution of the study participants ranged from 14 to 75 years old, with a mean of 34 years old, revealing significant insights into the demographic distribution of COVID-19 cases within the sample. Notably, 32% of participants were aged 21–30 years, followed by 21% in the 31–40 age group and 16% in the 41–50 age range. This age distribution suggests a predominance of younger individuals in the sample. This finding is aligned with a previous study conducted in Ethiopia by Sisay A. et al. [16]; a high number of study participants infected with SARS-CoV-2 belonged to the age group of 21 to 30. However, different findings by Varnica B. et al. [33] and Carmola L.R. et al. [34] reported that the elderly population was the most significantly afflicted in terms of the number of COVID-19 cases and deaths. These differences might be due to the predominance of younger individuals in our study sample, which has the potential for the underrepresentation of older populations.

The geographical distribution of participants, with the majority from Addis Ababa, highlights the urban concentration of cases, which is consistent with a study that analyzed COVID-19 infections in Zimbabwe and emphasizes that urban areas, characterized by condensed residences, tend to expedite SARS-CoV-2 transmission, leading to a higher total number of confirmed cases [36]. This is echoed by Sharifi and Khavarian-Garmsir [37], who discussed how urban planning and design can significantly influence the infection and mortality rates of COVID-19 and suggested that urban centers are more susceptible to outbreaks due to their structural characteristics. Moreover, they added that the dynamics of urban life, including increased mobility and social interactions, further exacerbate the spread of the virus.

The clinical presentation of symptoms such as cough, fever, and anosmia in 70% of participants underscores the typical manifestations associated with SARS-CoV-2 infections. This finding is consistent with the study by Elavia et al. [38]. They reported that fever was observed in approximately 88% of cases, while cough was presented in about 68% of infected individuals. They suggested that these symptoms are critical for clinicians to recognize, as they are often the first indicators prompting testing for SARS-CoV-2. Furthermore, similar findings demonstrated that fever and cough, along with anosmia, serve as independent predictors of positive SARS-CoV-2 PCR results, reinforcing their significance in clinical assessments [39]. This also aligns with previous findings that highlight the prevalence of these symptoms in COVID-19 patients, emphasizing their importance in clinical diagnosis and public health surveillance, as shown by Sisay A. et al. [16] and Tarris G. et al. [40].

We observed that 70 out of 150 samples had Ct values below 30 after retesting. This is particularly noteworthy, as lower Ct values are indicative of higher viral loads [41]. This study also identified, from 70 samples that had a cycle threshold (Ct) value of less than 30 and were sequenced, 63 samples passing bioinformatics quality control. This is supported by the study presented in [42] stressing the necessity of rigorous bioinformatics analysis to eliminate low-quality reads that could lead to erroneous conclusions regarding viral evolution. This suggests that the high pass rate of samples in this study highlights that the methodologies employed were robust and effective, which is critical for genomic surveillance efforts.

In this project, the sequencing analysis revealed a high-quality yield from the RNA extracts, with 99.8% of the sequences being of high quality and only 0.4% comprising gaps. This underscores the robustness of the sequencing methodologies employed, which align with the standards set by various genomic surveillance initiatives [26,43]. The genomic analysis of SARS-CoV-2 reveals significant insights into the virus’s mutation landscape and its evolutionary dynamics. This study revealed, on average, approximately 69 single nucleotide variations (SNVs) per genome, particularly concentrated in clades 22A and 21K, which is similar to the study conducted by Saldivar-Espinoza et al. [44] that explored the mutational landscape of SARS-CoV-2 for the Omicron variant by early June 2022 and reported that the average number of SNVs per genome was around 72. In addition, the study by Harvey et al. [45] and Yang et al. [46], highlighted that the identification of high mutation rates, particularly in certain lineages, suggests that these variants may be under selective pressure, possibly due to host immune responses or therapeutic interventions.

The findings from the Nextclade variant analysis identified a total of 4429 nucleotide mutations, 12,664 deletions, 4392 substitutions, and 84 insertions across 63 SARS-CoV-2 sequences compared to the Wuhan reference genome. This provides critical insights into the genomic evolution of the virus [26]. This extensive mutational landscape reflects the ongoing adaptive changes that SARS-CoV-2 undergoes in response to selective pressures, including host immune responses and antiviral treatments [47]. In this study, significant nucleotide mutations were identified in critical genes such as the spike (S), membrane (M), and nucleocapsid (N) genes, as well as non-structural proteins (nsp1, nsp6, nsp9, nsp10, nsp13, ORF3a, and ORF7a). Similar studies have corroborated these findings, highlighting the high mutation rates observed in SARS-CoV-2. For instance, a study conducted in Turkey reported elevated mutation rates in the S and RdRP gene regions, consistent with the findings of this analysis [26,48]. Furthermore, research from Egypt indicated a significant presence of the D614G mutation, which is known to enhance viral infectivity and has been widely documented across various global studies [49].

Moreover, this study identified specific mutations in the papain-like protease (PLpro) and the main protease (3CLpro). This study is aligned with the studies conducted by Gao et al. [50] and Ismail et al. [51] that suggested the mutations K38R, V1069I, and P132H in PLpro, along with P323L in RdRP, could potentially alter the enzymatic activities of these proteins, impacting viral replication and immune evasion.

The phylogenetic analysis performed in this study indicates that the dominant circulating variants in Ethiopia during the study period (June to August, 2022) were primarily Omicron, with a smaller proportion of Delta variants. The Omicron variant (B.1.1.529) of SARS-CoV-2 was first identified in November 2021 in South Africa [52]. It was classified as a VOC by the WHO because of its high transmissibility and evasion from neutralizing antibodies induced by vaccination or natural infection with the wild-type virus [53]. As of 14 April 2022, the Omicron variant has been confirmed in at least 150 countries around the world [54]. From 3 January 2022 to 9 January 2022, the Omicron variant accounted for roughly 62.5% of all SARS-CoV-2 sequences available on the GISAID database [25]. In this study, the majority of the samples belonged to the BA lineage. This is consistent with findings reported from North India, 2022 [55], and the Philippines, 2022 [56], during the study period. The Centers for Disease Control and Prevention (CDC) has developed taxonomic nomenclatures (BA, BF, AY, and B lineages) to categorize different variants of SARS-CoV-2 [57]. A study conducted by Rauseo AM et al. [58] suggests understanding the distribution of these variants is crucial for public health interventions, including vaccination strategies and treatment approaches.

This study explored the evolutionary divergence among genetic sequences of SARS-CoV-2 and revealed a spectrum of genetic distances ranging from nearly identical matches to slight variations. Specifically, comparisons within Ethiopian sequences revealed a perfect match between three samples and a very small distance between others, indicating recent community transmission. On a broader scale, the average genetic distance among Ethiopian sequences was low, suggesting high genetic similarity within the country. This finding is similar to a previous study conducted in Ethiopia through whole genome sequences of SARS-CoV-2 isolates from Ethiopian patients [59].

In addition, in this analysis, a comparison of the Ethiopian sequences with selected global sequences showed that a slightly higher average distance was observed. This indicates slightly more genetic diversity among global sequences. These findings were contrary to the evolution of SARS-CoV-2 being characterized by a steady increase in divergence within major lineages and a stepwise increase associated with each successive new major lineage, leading to a faster overall rate of evolution [60]. However, another study that analyzed 27,977 SARS-CoV-2 sequences from 84 countries obtained throughout the pandemic showed that SARS-CoV-2 genetic diversity was remarkably low [61].

### Limitations of This Study

One limitation of our study is the small sample size that limits the power to give more conclusive recommendations. Another limitation is geographical coverage. Due to financial and infrastructure limitations, the sequencing was conducted very late; however, the data are still novel for readers who are interested in the SARS-CoV-2 variant situation in Ethiopia.

## 5. Conclusions

In conclusion, this study on SARS-CoV-2 variants in Ethiopia during the fifth wave reveals critical insights into the demographics, clinical characteristics, and genomic evolution of the virus, highlighting the predominance of the Omicron variant, particularly sub-lineages BA.4.1 and BA.4.1.1. The findings indicate a significant male predominance among participants, consistent with global trends of higher severity in males, and underscore the typical clinical manifestations of COVID-19, such as cough, fever, and anosmia. The genomic analysis suggests minimal genetic divergence among local sequences, indicating stable transmission dynamics, while the presence of specific mutations raises concerns regarding vaccine efficacy and potential immune evasion. These results emphasize the necessity for ongoing genomic surveillance to monitor variant emergence and inform public health strategies, contributing to a broader understanding of the pandemic’s trajectory in Ethiopia and its implications for global health responses.

## Figures and Tables

**Figure 1 genes-16-00351-f001:**
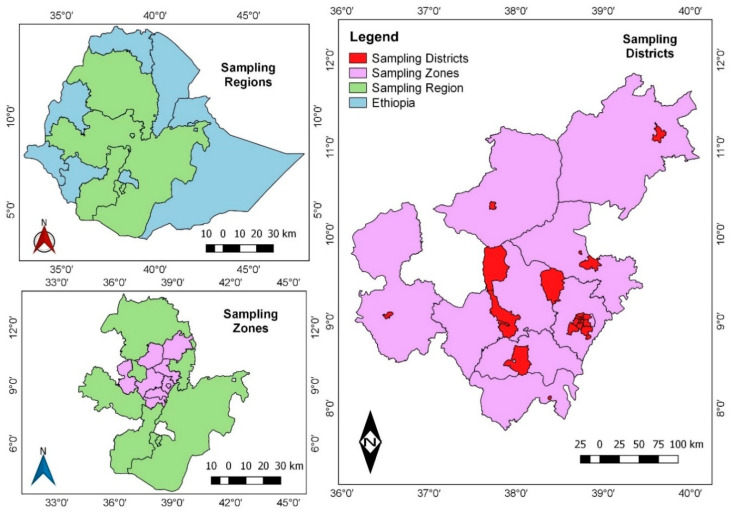
A geographic map of Ethiopia showing the regions and districts where SARS-CoV-2 specimens were collected.

**Figure 2 genes-16-00351-f002:**
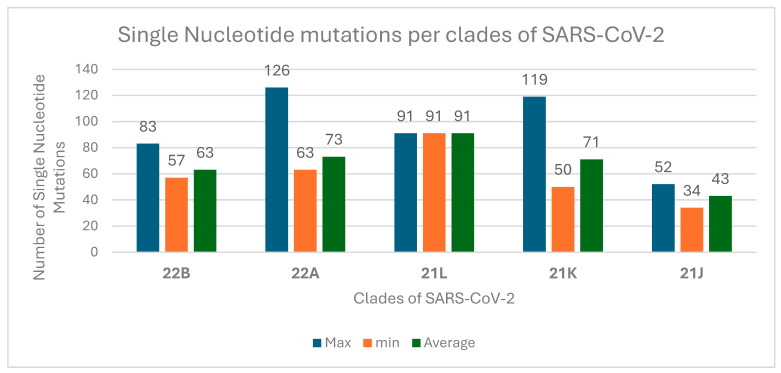
Number of single nucleotide mutations per clades of SARS-CoV-2 during the fifth wave of the pandemic in Ethiopia, 2022.

**Figure 3 genes-16-00351-f003:**
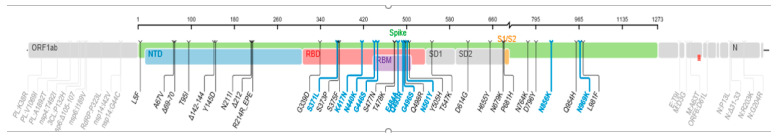
Example of nucleotide mutations in different gene positions of SARS-CoV-2 during the fifth wave of the pandemic in Ethiopia, 2022.

**Figure 4 genes-16-00351-f004:**
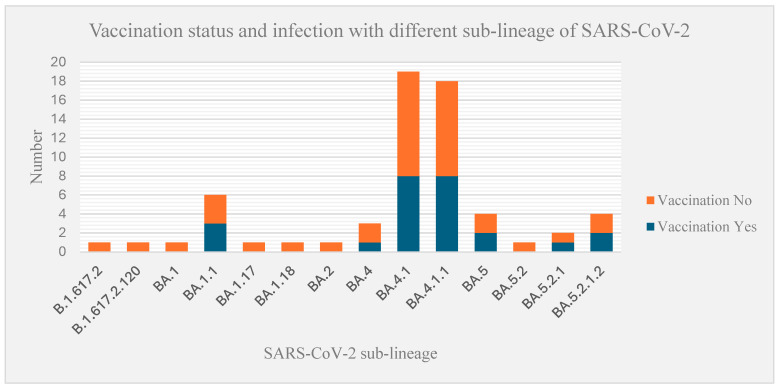
SARS-CoV-2 vaccination status and infection with different sub-lineages of SARS-CoV-2 identified during the fifth wave of the pandemic in Ethiopia.

**Figure 5 genes-16-00351-f005:**
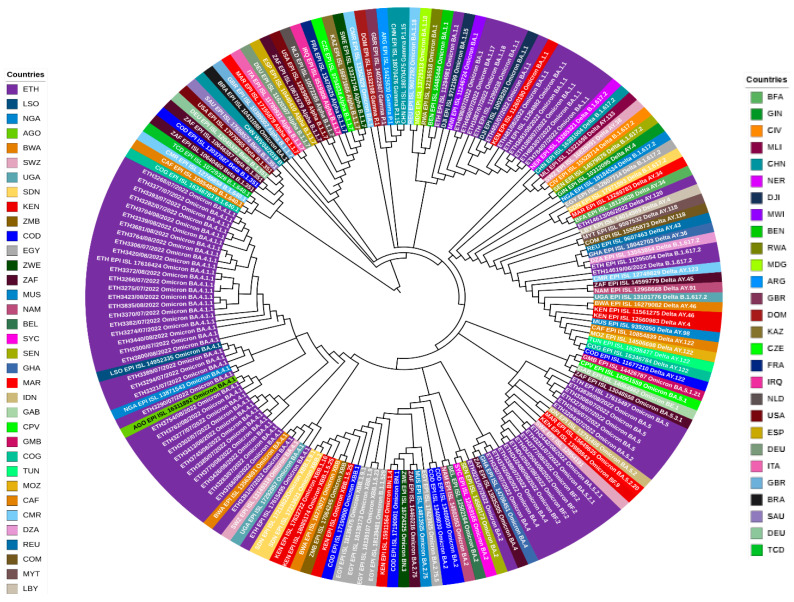
Analysis of the phylogenetic relatedness of SARS-CoV-2 identified in Ethiopia during the fifth wave of the pandemic, 2022, with selected global sequences.

**Table 1 genes-16-00351-t001:** The WIV04 genome, along with 118 SARS-CoV-2 genome sequences from 62 countries, were obtained from the GISAID.

Classifications	Variants	Name	First Reported Country
VOC	alpha	GRY (B.1.1.7 + Q.*)	United Kingdom
VOC	Beta	GH/501Y.V2 (B.1.351 + B.1.351.2 + B.1.351 + 3)	South Africa
VOC	Gamma	GR/501Y.V3 (P.1 + P.1*)	Brazil and Japan
VOC	Omicron	GRA (B.1.1.529 + BA.*)	Botswana, South Africa, and Hong Kong
Variant of interest (VOI)	Lambda	GR/452Q.V1 (C.37 + C.37.1)	Peru
VOI	Mu	GH (B.1.621 + B.1.621.1)	Colombia
VOI		GRA (XBB.1.5 + XBB.1.5.*)	Australia, India, and Bangladesh
VOC		GRA (XBB.1.16 + XBB.1.1.16.*)	India
Variant under monitoring (VUM)		GRA (XBB + XBB.*, excluding XBB.1.5, XBB.1.16, XBB.1.9.1, XBB.1.9.2, XBB.2.3)	India
VUM		GRA (XBB.1.9.1 + XBB.19.1.*)	Indonesia, Singapore, and Israel
VUM		GH/490R (B.1.640 + B.1.640.*)	Congo and France. Except for the Gamma VOC

Note: These were sequences that we collected between 1 January and 31 July 2022 before our samples were collected between June and August 2022. There was a total of 30,150 positions in the final dataset.

**Table 2 genes-16-00351-t002:** Socio-demographic characteristics of study participants.

Characteristics		Number (n)	Percentage (%)
Sex	Female	19	30
	Male	44	70
Age Group	≤20	8	13
	21–30	20	32
	31–40	13	21
	41–50	10	16
	51–60	4	6
	>60	8	13
Testing Sites	Oromia	15	24
	Amhara	3	5
	Addis Ababa	44	70
	SNNPR	1	1
Signs and Symptoms	No	19	30
	Yes	44	70
Co-morbidity	No	36	57
	Yes	27	43
Travel History	No	55	87
	Yes	8	13
Vaccination Status	Not vaccinated	38	60
	Vaccinated	25	40
Reason for Testing	Contact with cases	10	16
	Follow-up	2	3
	Suspect	43	68
	Travel purpose	8	13

**Table 3 genes-16-00351-t003:** Frequency distribution of SARS-CoV-2 variants with demographic and clinical information of patients during the fifth wave of the pandemic in Ethiopia, 2022.

			Variants of SARS-CoV-2 Virus	
			Delta	Omicron
Sex	Female	Number	1	18
		%	5.3	94.7
	Male	Number	1	43
		%	2.3	97.7
Vaccination	No	Number	2	36
		%	5.3	94.7
	Yes	Number	0	25
		%	0.0	100.0
Signs and Symptoms	No	Number	0	19
		%	0.0	100.0
	Yes	Number	2	42
		%	4.5	95.5
Travel History	No	Number	2	53
		%	3.6	96.4
	Yes	Number	0	8
		%	0.0	100.0
Ethiopian Nationality	No	Number	0	1
		%	0.0	100.0
	Yes	Number	2	60
		%	3.2	96.8
Age Group	≤20	Number	0	8
		%	0.0	100.0
	21–30	Number	1	19
		%	5.0	95.0
	31–40	Number	0	13
		%	0.0	100.0
	41–50	Number	1	9
		%	10.0	90.0
	51–60	Number	0	4
		%	0.0	100.0
	>60	Number	0	8
		%	0.0	100.0

## Data Availability

The sequence data are available at GISAID and NCBI.

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
