# Peer review of "SARS-CoV-2 Genetic Variants Identified in Selected Regions of Ethiopia Through Whole Genome Sequencing: Insights from the Fifth Wave of COVID-19"

_genes, 2025, doi:10.3390/genes16030351_

Round 1
Reviewer 1 Report
Comments and Suggestions for Authors
This is a well written summary of a portion of the COVID pandemic in Ethiopia. It is a useful contribution to the scientific literature and should be welcome to a specialist readership.
Note:
define all waves by start and end date in Ethiopia. You need a list of when Wave 1 started and ended, when Wave 2 started and ended, and so on. Not all readers will know wha "Wave 5" is. Put this in the Introduction.
ABSTRACT
Remove from Abstract
"Samples were re-analyzed using the BIO-RAD CFX96 Deep Well™ Real-Time PCR system. Seventy samples with Ct values <30 underwent next-generation sequencing on the Illumina NextSeq 550 platform."
this belongs in Methods. It is not important enough to be in Abstract
"Sixty-three high-quality genome sequences were analyzed using bioinformatics tools, with the reference genome "hCoV-19/Wuhan/WIV04/2019" used for phylogeny."
just write
"Sixty-three high-quality genome sequences were analyzed"
for the Abstract. Put the rest of the information in Methods
"Additionally, 118 sequences from GISAID, representing 18 neighboring and global regions, were included to assess evolutionary divergence."
remove from Abstract. put in Methods
"Results: Of the 63 sequences, 69.8% were from males and 30.2% from females, with a median age of 34."
too much precision. just write
"Results: Of the 63 sequences, 70% were from males and 30% from females, with a median age of 34."
MAIN BODY
"United Kingdome"
"georphical"
correct spelling
use a spell checker! read your manuscript carefully!
"The evolutionary divergence between sequences ranges from 0.00 to 3.45 × 10−3."
add units to these numbers
Also check throughout the paper for any reported numbers that do not have their units afterwards. There are several other instances
Figure 5 does not work
Mainly because many terminal branches are not labeled. These need to be labeled.
It is not clear that you need to include all the terminal sequences in order to make your point. Maybe eliminate about 40% of the sequences used for this dendrogram, and use it to illustrate the point you want. Folks can always generate this graph online if they need all the details. They don't need it in a pdf.
Figure 6 also does not work. Too much white space. Either refactor the legend with all the countries so it is not vertical, or maybe lump some countries together. E.g. "Europe". Or delete some reference sequences that do not add value.
"This study identified high number (70%, 44/63) of males were infected with SARSCoV-2 during the study period."
delete this sentence. You didn't really design this study to address whether the number of infected males was high. I assume you mean "higher than women". Because obviously a lot of people, including males, were infected. You don't really need to say this. And if you mean that more men than women were infected, then you need to spend a lot more effort in the paper describing how you completely unbiasedly sampled the Ethiopian population. Something that woudl seem exceptionally hard to do.
"Data Availability Statement: The original contributions presented in this study are included in the article. Further inquiries can be directed to the corresponding authors."
Not sufficient. Delete the part about "Further inquiries can be directed to the corresponding authors." This no longer works. It was used in the 20th century but is not OK in the 21st. And delete "The original contributions presented in this study are included in the article." as that is obvious. Just state where you deposited the sequences. Ideally you deposited them to both NCBI and GSAID. Can you confirm that these sequences are in both NCBI and GSAID?
Author Response
Dear Reviewer,
Thank you for your time and effort in improving our manuscript. I sincerely appreciate your constructive and supportive comments and suggestions. I have addressed all the comments and attached my responses here.
Best regards,
Getnet Hailu

Reviewer 2 Report
Comments and Suggestions for Authors
The manuscript submitted for review with the title “SARS-CoV-2 genetic variants identified in selected regions of Ethiopia through whole genome sequencing: Insights from the fifth wave of COVID-19” describes a retrospective sequencing study of the fifth wave of SARS-CoV-2 infection in Ethiopia. Socio-demographic data are also presented.
The iThenticate report shows Percent match: 77%. The high percentage of identity is due to the fact that the manuscript have been uploaded to Research Square.
Authors should follow the journal’s instructions for manuscript preparation: In the “Abstract”, subheadings should be bolded; citations in the text should be enclosed in [] brackets; the references after the text should be prepared according to the journal’s requirements. The same should be done for the main text.
The abstract should be revised (there is redundant text) and I have some recommendations related to this: the abbreviation EPHI should be removed; in the text “...... with the reference genome "hCoV-19/Wuan/ WIV04/2019" used for phylogeny.” – this part of the sentence is redundant. The authors present results related to vaccination status, mortality, travel history and other characteristics presented in the “Results” section, Table 2. There is no connection or conclusion in the Abstract regarding these data and the results of the sequences.
Introduction: The “Introduction” is informative, providing information about the development of the pandemic in Africa and Ethiopia. The “Introduction” provides a justification for why this study is being conducted now and why it is retrospective. I have the following recommendation: The following sentence is not clear whether it is about cities or…: “A seroepidemiological survey conducted in Addis Ababa and Jimma between August 2020 and April 2022 revealed that over 96% of the study group had been exposed to SARS-CoV-2 at least once”.
The “Material and methods” are described in detail, and the manufacturers' instructions are followed everywhere. I have the following recommendation: Figure 1 is not very clear to me - I do not distinguish between the country and the individual regions. Therefore, it would be nice to label the individual images with A, B, C, and for Sampling Zones or Ethiopia to choose a different color (the colors used now are very close, and a large percentage of people have reduced color perception).
The results are well presented. I have the following recommendation: table 3 can be moved as a supplementary file. Figures 5 and 6 are missing the bootstrap values ​​from which conclusions can be drawn about the phylogenetic relationship between the sequences. The authors have to add and discuss this data.
Discussion: I have some recommendations related to this part of the manuscript. In the “Discussion” there are statements from the authors that need evidence. Also, individual statements need additional information. In this regard, for the convenience of the authors in certain places I make suggestions. For example: “The virus is primarily transmitted through respiratory droplets and direct contact.” – need citation. Also, the spread of the virus depends on various factors, which can be added to the above sentence – sunshine, climate, temperature, humidity, population demographics, etc. (https://doi.org/10.22207/JPAM.18.1.27). Also, asymptomatic and pre-symptomatic infected people are a key factor in the spread of SARS-CoV-2 (examples: Sirakov I, Stankova P…... Asymptomatic spread of SARS-CoV-2 during the first wave in Bulgaria: A retrospective study in a region with distinct geography and climate. Was the virus source from the UK. Acta Microbiol. Bulg. 2022;38:358-60; Zhang L, Zhang Z, Pei S, Gao Q, Chen W. Quantifying the presymptomatic transmission of COVID-19 in the USA. Mathematical Biosciences and Engineering. 2024;21(1):861-83).
„It possesses a large RNA genome that encodes various proteins essential for its lifecycle and pathogenesis, with the spike protein being particularly important for viral entry and a key target for neutralization. Mutations in the virus can lead to the emergence of new variants that may impact transmission rates, disease severity, and vaccine efficacy“ (Citation missing. Example: https://doi.org/10.24321/0019.5138.202232 ).
“In Ethiopia, the first COVID-19 case was reported in March 2020, and by April 2024, the country had recorded over 500,000 cases, with approximately 38% of the population vaccinated (need citation). A seroepidemiological survey indicated high exposure rates among the population (need citation).”
“In addition, Li et al.(39) reported that the infection risk of SARS-CoV-2 may not significantly correlate with sex, age, or race, although mortality risks are higher for males. Then they highlighted the complexity of understanding SARS-CoV-2 transmission dynamics and the need for nuanced interpretations of gender-related data.”- The authors cite a statement by other researchers. How do the authors interpret their results in the context of the above sentence? The authors need to present their point of view based on their results and compare them with others, even those that contradict theirs, and analyze them. The same is true for the previous sentence in the manuscript “However, another study that analyzed data from 1,019 COVID-19 patients found that 50.0% of the infected individuals were male”.
“The clinical presentation of symptoms such as cough, fever, and anosmia in 70% of participants underscores the typical manifestations associated with SARS-CoV-2 infections.” The “typical manifestations associated with SARS-CoV-2 infections” –This has been established by other authors before. A citation is needed. In the same paragraph, it is again described what other researchers have done and what they think. All this happens by the end of the “Discussion”. Please correct. Authors should state their opinion and statement based on their data and other studies. When their results differ from those of other researchers, then the authors should explain, in their opinion, why this is so.
The authors present results related to vaccination status, mortality, travel history and other characteristics presented in the Results section, Table 2. There is no discussion regarding these data and the sequencing results.
What is the duration of the fifth wave in Ethiopia? Why were these three months chosen for sampling? – These questions should be discussed in the text.
In conclusion – The “Discussion” part should be optimized according to the guidelines above – remove the descriptive part of what others have done and replace it with an analysis and discussion based on your data and other people's results and conclusions.
There are many published studies from Ethiopia that may be useful for the authors to explain the phylogenetic results and socio-demographic data, which can subsequently be linked to the epidemiology of SARS-CoV-2.
I am not qualified to assess the quality and level of the English language. The platform requires me to give an assessment that editors and authors should not comply with.
Author Response
Dear Reviewer,
Thank you for your time and effort in improving our manuscript. I sincerely appreciate your constructive and supportive comments and suggestions. I have addressed all the comments and attached my responses here.
Best regards,
